

# The Subduction Dichotomy of Strong Plates and Weak Slabs

Robert I. Petersen[1], Dave R. Stegman[1], and Paul J. Tackley[2]

[1]Institute of Geophysics and Planetary Physics University of California, San Diego 9500 Gilman Drive, La Jolla CA 92093-0225, USA.
[2]Institute für Geophysik ETH Zürich Sonneggstrasse 5, 8092 Zürich, Switzerland.

*Correspondence to:* Robert I. Petersen (rpetersen@ucsd.edu)

**Abstract.** A key element of plate tectonics on Earth is that the lithosphere is subducting into the mantle. Subduction results from forces that bend and pull the lithosphere into the interior of the Earth. Once subducted, lithospheric slabs are further modified by dynamic forces in the mantle and their sinking is inhibited by the increase in viscosity of the lower mantle. These forces are resisted by the material strength of the lithosphere. Using geodynamic models we investigate several subduction

models wherein we control material strength by setting a maximum viscosity for the surface plates and the subducted slabs independently. We find that the models which produce results most analogous to observations of subduction on Earth are characterized by a dichotomy of lithosphere strengths. These models have strong lithospheric plates at the surface which promotes Earth-like single-sided subduction. At the same time these models have weakened lithospheric subducted slabs which pile, bend or lie flat at the top of the lower mantle reproducing the spectrum of slab morphologies observed on Earth.

## 1 Introduction

A key element of plate tectonics is the recycling of lithospheric plates into the mantle. Colder and more dense slabs, already having been subducted, generate the driving force that pulls and bends tectonic plates below the surface. Coupling of tectonic plates at the surface, shear stress on the subducted plate, induction of mantle flow around the subducting plate, the contact of slabs with upper mantle/lower mantle boundary and material strength of plates generate resisting forces which resist bending

inhibit subduction. The particular form and speed of subduction is controlled by the balance of the opposing forces (Jarrard, 1986; Royden and Husson, 2009; Petersen et al., 2015).

On Earth this process is asymmetric, occurring at convergent boundaries where one of the two plates is subducted while the other plate, the overriding plate, remains at the surface. The mechanical strength of lithospheric material needs to be sufficiently strong so that the overriding plate can resist bending while weak enough so that the subducting plate can bend and subduct.

Subducted slabs moving through the mantle and encountering the higher viscosity lower mantle experience forces that deform the slab. These forces are transmitted to the surface, the slab acting as a stress guide, affecting the state of stress at the trench and the coupling of subducting and overriding plate (Clark et al., 2008).

Models produce a diverse set of mantle convection styles (Solomatov and Moresi, 1997; Crameri et al., 2012b; Gerya et al., 2008; O'Neill, 2012; Lenardic and Crowley, 2012; O'Rourke and Korenaga, 2012), including stagnant lid, 2-sided down-

wellings, and 1-sided subduction. Our geodynamic models use the rheological laws of visco-plastic, temperature-dependent



material to provide a laboratory for examining subduction systems. By introducing a temperature and depth dependency on the maximum viscosity allowable we are able to investigate subduction where the cold material of subducted slab is weaker than the surface plate. We can address questions regarding the resistance of surface plates to bending and the morphology of subducted slabs in the upper mantle. Our models allow for contrasting viscosity at depth enabling us to look at how the

presence or absence of a higher viscosity lower mantle affects the system.

## 2   Methods

We develop two-dimensional models of convective systems using the finite volume code, StagYY (Tackley, 2008). StagYY is a numerical model of solid-state mantle convection that solves the conservation equations for energy, momentum, and mass. The models use an incompressible material with an infinite Prandtl number approximation.

Dimensional parameters are used and parameters common to all models are given in Table 1. A schematic of the model setup is shown in Figure 1. The aspect ratio for all models is 4 to 1, with dimensional length of 5440 km and depth of 1360 km. In this study we examine models both with and without a lower mantle. For models with a lower mantle, the domain is divided into upper and lower portions with a size of 660 km and 600 km respectively. The viscosity of the lower mantle is 50 times that of the upper mantle. A 100 km layer of sticky-air (Schmeling et al., 2008; Crameri et al., 2012a, b)is placed at the top

of the model. The model space is gridded using a grid of $1024 \times 256$ grid points, with vertical refinement around the air/rock interface. The minimum vertical resolution is 1.5 km and the maximum is 5.9 km.

### 2.1   Rheology

We use a strongly temperature-dependent Arrhenius relation to calculate the viscosity of a silicate mantle, show in equation 1.

$$\eta(T) = Ae^{\frac{E}{RT}} \tag{1}$$

where $\eta(T)$ is the temperature-dependent viscosity, $R$ is the gas constant, $T$ is temperature and $E$ is the activation energy ($240\mathrm{kJmol}^{-1}$). The prefactor $A$ is calculated so that a reference mantle temperature of $T_0 = 1600\mathrm{K}$ results in a reference viscosity $\eta_{ref} = 1 \times 10^{20}\mathrm{Pas}$. If the temperature-determined viscosity falls outside a specified window, a cut-off is applied. In this study the maximum allowed viscosity, $\eta_{max}$ is one of $10^{23}$, $10^{24}$ or $10^{25}$ Pas, and the minimum viscosity of non-air material is $\eta_{min} = 10^{19}\mathrm{Pas}$. Weakening of the subducted slab is accomplished by lowering the maximum viscosity at depth for material

below 1000 K. In the models where slab weakening is used, material 10 km below the depth of the lithosphere/asthenosphere boundary is assigned a lower maximum viscosity than the surface plates. We use plates of 50 km and 80 km in this study and the weakening is applied to material below 60 km and 90 km respectively. We examine cases where the weakening is 1%, 10%, or 100% (no weakening) of the maximum viscosity cut-off of the surface plates. The method of lowering the maximum viscosity at depth is a computational efficient way of parameterizing the effective viscosity that may result due to any number

of mechanisms such as Peierls creep, damage rheology or other Non-Newtonian rheologies (Kameyama et al., 1999; Bercovici, 2003; Garel et al., 2014; Bercovici et al., 2015; Holt et al., 2015a).



Material is subject to failure through an applied yield strength failure criterion. The yield stress, $\sigma_{yield}$, follows Byerlee's law and is pressure dependent for the mantle,

$$\sigma_{yield} = C + p\mu \qquad (2)$$

where $C$ is the cohesion, $\mu$ is the coefficient of friction, and $p$ is the hydrostatic pressure. For the bulk of the mantle $C = 150.0\,\mathrm{MPa}$, $\mu = 0.1$. For the weak crust layer $C = 1.0\,\mathrm{MPa}$, $\mu = 0.0$.

## 2.2 Initial condition

A temperature field and particle cloud are the initial conditions for all models. The temperature field is defined by a uniform background temperature of 1600 K and plates with a half-space cooling temperature gradient. The viscosity of the background and plates is calculated using equation 1, subject to cutoffs.

The surface of each model is comprised of two plates. The initial downgoing plate begins 20 km from the left wall. It extends to the center of the box (the initial trench) and then follows a path with radius of curvature 400 km into the mantle to a depth of 200 km.

The overriding plate occupies the right side of the box, from 20 km right of the trench to 200 km short of the right wall. The boundary conditions are reflecting along the sides of the model domain, free slip at the top and no slip along the bottom.

## 2.3 The surface

These models use a pseudo free surface in the form of a low density layer at the top of the model called sticky air (Schmeling et al., 2008). The viscosity of the air is $10^{18}\,\mathrm{Pa\,s}$, ten times lower than the minimum allowed viscosity of the mantle material.

To promote 1-sided subduction (Crameri et al., 2012b) the top 8 km of the mantle material is a layer of weak crust, which has a lower yield strength as described above. The material accommodates the plastic deformation of the bending plate (Capitanio et al., 2009).

## 2.4 Radius of Curvature

Radius of curvature has been used in several studies as a metric of subduction (Becker et al., 1999; Conrad and Hager, 1999; Buffett and Heuret, 2011; Holt et al., 2015b). In this study we first examine three methods for calculating the radius of curvature. All methods use tracer particles that represent the crust of the subducted slab.

The "spline" method has been used previously in studies to calculate a minimum radius of curvature for subducting slabs in both numerical models and in the Earth (Buffett and Heuret, 2011; Holt et al., 2015b). This method fits a smooth cubic spline to points, or observed locations of earthquake hypocenters (Buffett and Heuret, 2011), with a penalty on the second derivative (De Boor, 1978). The size of penalty is controlled by a weighting parameter between 0 and 1. When the parameter is 0, the method returns a least square straight line fit to the data. When it is 1, the method returns a piece-wise cubic spline passing through the data. As input we use crustal particles along the subducted slab from the trench to the 150 km depth.





In a second method, which we call the "angle" method, we first fit a curve to the base of the crust. We then select crust particles from the trench to the point where that curve has a slope of -1. Using a least squares fit we fit a circle to the crustal particles. The third method, the "depth" method, fits a circle as in the "angle" method using crustal particles from the trench to a depth of 150 km.

## 3 Results

### 3.1 Model Evolution

We ran 36 models varying the plate strength and weakening parameter of slabs (full list of parameters in Table 2). The evolution of all models is comparable in terms of their general behavior and slab morphology. All models begin with the slab tip sinking and the radius of curvature adjusting from the initial condition towards a dynamic equilibrium.

For models with a lower mantle, the slab descends through the upper mantle and encounters the boundary between the upper and lower mantle. The descent of the slab is inhibited at this boundary. Lateral motion of the slab tip is also inhibited either because the tip penetrates into the lower mantle, or because the slab rests on the boundary and is subject to shear traction between the slab and lower mantle. The slab then lies flat on the upper/lower mantle boundary and then begins buckles and pile. Eventually the piled slab descends through the boundary as a unit, see Figure 2.

For models with no lower mantle the slab descends until it encounters the bottom of the model box. The bottom has a no-slip boundary condition. When the slab encounters the bottom of the box it lies down and buckles in a manner similar to those models with a lower mantle in which the slab becomes embedded.

### 3.2 Radius of curvature

We calculated the radius of curvature for every timestep, every model using the three methods described above. The initial condition imposed a radius of curvature of 400 km. In the $0^{th}$ timestep no method returned 400 km. The "depth" method always returned the closest radius of curvature. The average across models for the $0^{th}$ timestep for "depth", "angle", and "spline" methods was 370 km, 337 km, and 113 km, respectively.

The three methods of calculating the radius of curvature for subducted slabs return differing values for any given time step in the model. Figure 3 shows the results for a representative model (14), at a single timestep. The figure shows the crustal particles, downsampled to enhance visibility, in gray. For each of the least squares circle fit method, "angle" (red) and "depth" (blue), the calculated circle is plotted using center point and radius returned by the method. The smoothed spline (magenta) is shown on top of the slab particle dots. The green circle shows the minimum radius of curvature given by the spline method. The circle is plotted to be tangent to the spline at the point of minimum radius of curvature and with the appropriate radius. Figure 4 shows the radius of curvature calculations for the same model over time. For each of the three methods, the relative size of the resultant radii of curvature changes during the model run. The first movement of the slabs is due to the slabs relaxing from the initial state and is characterized by a decrease in radii of curvature. The radii of curvature continue to decrease as the slabs descend



through the upper mantle. During the initial period of relaxation from the initial condition the "depth" method calculates a larger radius of curvature than the "angle" method which is larger still than the "spline" method. Upon encountering the lower mantle both "circle" methods calculate a similar radius of curvature and increase the rate of radius of curvature reduction. The "spline" method returns roughly equivalent values with during the relaxation period and during the period in which the slab

encounters the lower mantle. The minimum radius of curvature returned by all methods happens during the period when the slab is resting on the upper mantle. During the phase of buckling and piling the radius of curvature for the two "circle" methods rise and fall, while the "spline" method becomes increasingly noisy. For the purpose of model intercomparison we selected the "depth" method to calculate radius of curvature.

Figure 5 shows the time evolution of the radius of curvature calculated using the "depth" method for two models, 18 and 36,

identical in model setup except for the presence of a lower mantle in 18. The radius of curvature for both models is more or less the same during the period of relaxation. At about 5 million years in model time they begin to diverge. The radius of curvature for model 36, with no lower mantle, rapidly decreases until about 6 million years with the slab encounters the bottom of the box. The radius of curvature for model 18, continues to decrease at about the same rate.

Figure 6 shows the radius of curvature over time for three models with varying maximum viscosities of the surface plate. The

blue (model 17, $10^{25}$ Pa s), green (model 14, $10^{24}$ Pa s), and red (model 11, $10^{23}$ Pa s), plots show the strong, intermediate, and weak plates. The slabs all have the same amount of relative weakening, a maximum slab viscosity of 10% of the plate viscosity. The inset in the plot shows the 1550 K isotherm, with same color scheme as the plot at a model time of 6 million years. The plot shows weaker plates have a larger rate of radius of curvature reduction. As the radius of curvature changes due to the buckling of the plates (~7-11 Ma for model 11, ~12-15 Ma for 14, and ~16-19 Ma for 17), the maximum radius of

curvature during that period is highest for the strongest plate, intermediate for the intermediate strength plate, and lowest for the weakest plate.

Figure 7 shows the radius of curvature during model runs for three models with the same plate strength, $10^{24}$ Pa s, and differing slab strengths (model 13 1%, model 14 10%, model 15 100%). In these models the decrease in the radius of curvature is fastest for the weakest slab models and slowest for the strongest slab model. During the period of buckling the maximum

radius of each models is within 50 km of each other. The length of this phase is the longest for the weakest slab at almost 10 Ma, while for the strongest slab the phase lasts ~3 Ma, and for the intermediate slab the phase is about 6 Ma.

In Figures 8 to 11 we compare the varying slab morphologies emergent in models of common plate strength and slab weakening.

Figure 8 compares models with the strongest plate strength tested in this study, plates with a maximum viscosity of $10^{25}$ Pa

s and a lithospheric thickness of 80 km. The three models, 16, 17, and 18, have slab weakening parameters of 1%, 10%, and 100%, respectively. The two stronger slab models, 17 and 18, penetrate the lower mantle. Model 18 penetrates the deepest and bends allowing the slab to lay flat at the top of the lower mantle. Model 17 does not penetrate as deep and in laying flat does not bend the portion of the slab which has just entered the mantle. The weakest slab was deflected as it approached the lower mantle and the tip remains in the upper mantle.





A similar comparison of models of equal plate strength and varied slab strength is shown in Figure 9. In this figure we plot the weakest plates of the model suite. The plates are thinner (50 km) than those discussed above and have a lower plate maximum viscosity, $10^{23}$ Pa s. These plates are easily deflected as they approach the lower mantle and the tips do not penetrate. The tips are deflected upward as the following plate descends. The result is a shillelagh shaped slab tip. Model 3, at the timestep

shown, has begun the convective style of a two-sided downwelling as the overriding plate is not strong enough to resist the downward forces at the plate interface.

In Figures 10 and 11 we show comparisons of models with varying plate strength and equal slab strength. Models with 80 km thick plates, slab maximum viscosity of $10^{23}$ Pa s, and plate viscosities of $10^{23}$ Pa s (model 12), $10^{24}$ Pa s (model 14), and $10^{25}$ Pa s (model 16), are shown in Figure 10. The relatively weak slabs lie at the bottom of the upper mantle and have

a similar shape at depth. The shape of the subducted slabs found in the shallow mantle are differentiated by their radius of curvature. Model 16, with the strongest plate, has the largest radius of curvature while the model 2, with the weakest plate, has the smallest radius of curvature of the three models.

Figure 11 shows a set of models with weaker surface plates than those shown in Figure 10. The maximum viscosities of the plates are $10^{23}$ Pa s (model 3), $10^{24}$ Pa s (model 5), and $10^{25}$ Pa s (model 7) but the thickness of the plates has been reduced

to 50km. These models have smaller radii of curvature and the slabs approach the lower mantle at a less acute angle than their 80 km thick plate counterparts. As such the tips of the slab in these models are deflected more and, like the models plotted in Figure 9, take on a shillelagh shape. Here again we see that the subducting plate of model 3 has begun to pull down the overriding plate.

## 4 Discussion

### 4.1 Radius of Curvature

The observed states of models, relaxing from the initial condition, encountering the upper/lower mantle boundary and buckling are associated with changes to the radius of curvature as calculated by the two circle methods. The "spline" method for calculating radius of curvature is not sensitive to these states. The "depth" method captures a greater number of crust particles than the "angle" method and is consequently less noisy. Noisier still is the "spline" method. The "spline" method is subject

to an arbitrary smoothing penalty. The minimum radius of curvature is a function of far fewer points than either of the circle methods. The points closest to the point of minimum radius of curvature are fit smoothly and have the largest effect on the radius of curvature while points at increasing distance have less weight.

Radii of curvature are commonly used to calculate bending dissipation which is balanced against the gravitational potential energy of sinking slabs (Conrad and Hager, 1999; Buffett and Rowley, 2006; Buffett and Heuret, 2011; Holt et al., 2015b).

Numerical models are advantaged in that the energy of bending dissipation can be calculated within the model and subject to full set of dynamic forces which are incorporated therein. Further, observations of earthquakes used to calculate the radius of curvature are limited in number as compared to numerical models which can employ many more tracers to which a particular method can be fit against. Finally, the calculations of radius of curvature are limited to the present day for observations of





Earth, whilst models provide a time series of radii of curvature. Studies that use earthquake hypocenter locations Wu et al. (2008); Buffett and Heuret (2011) and the spline method likely underestimate the radius of curvature and calculate dissipation for a single time step invoking "steady state" subduction, the appropriateness of which is discussed below.

## 4.2 Strong Plates

The steeply dipping slabs apply a stronger torque on the overriding plate than shallow dipping slabs. Experiments, numerical and analog, suggest that tectonic plates are strong, with viscosity contrasts several hundred times that of the mantle (Gerya et al., 2008; Capitanio et al., 2010; Stegman et al., 2010b, a). In our experiments, strong subducting plates maintain their large radius of curvature for a longer time period than their weaker counterparts. Weaker plates quickly relax from the initial condition. Weak plate models more quickly transition from singled-sided subduction to two-sided downwellings. This pathway
to two-sided downwellings is discussed at length in Petersen et al. (2015). Models in Petersen et al. (2015) typically had plates with strengths similar to the weakest plates in this study and also explored weaker plates in which the transition to two-sided downwellings occurs almost immediately. Model features like strong plates with a weak coupling at the plate interface, such as a plate boundary fault, suppressed feedback which resulted in two-sided downwellings.

## 4.3 Weak Slabs

Numerical models and observation reveal a rich variety of subducted slab morphologies. Computational models may modify plate strength and rheological laws to a variety of subduction styles (Stegman et al., 2010b, a; Garel et al., 2014; Petersen et al., 2015). Subducted slabs may pile at the upper/lower mantle boundary as suggested of the Sunda and Farallon slabs (Ribe et al., 2007), may penetrate in the lower mantle (Zhong and Gurnis, 1995) and may lie flat along the boundary (Fukao et al., 2001). Other work has suggested that slabs stalled at the upper/lower mantle boundary may eventually "avalanche" into the lower
mantle (Stein and Hofmann, 1994; Breuer and Spohn, 1995; Condie, 1995). The models of this paper with strong plates and strong slabs result in some models where the slab tip is embedded into the lower mantle as the slab descends through the upper mantle. Once embedded there is no lateral motion of the subducted slab and it begins to buckle. The stresses are transmitted to the surface and modify the radius of curvature over a short period of time. The very weakest slabs do not immediately penetrate into the lower mantle, rather they rest on the boundary. Weaker slabs exhibit piling and flat lying slabs and represent the most
analogous behavior in regards to the range of observed slab morphologies in Earth. The stresses that bend the subducted slab to pile it are weaker and do not modify the radius of curvature as rapidly. Eventually these slabs do descend into the lower mantle.

## 4.4 No Lower Mantle

The effects of a lower mantle on subduction and the extent to which the lower mantle participates in convection are debated
on both geophysical and geochemical grounds (Davies, 1977; Loper, 1985; van Keken and Ballentine, 1998; Tackley, 2000; Stegman et al., 2002). In this study we constructed models with a lower mantle and without, both of which had the same vertical





extent. Our models find that the absence of a lower mantle has the effect of speeding up subduction due to the higher Ra as compared to models with a lower mantle. The lower boundary of the model has a no slip boundary condition. As such strong plates behave similarly as when the lower mantle is present, i.e. the horizontal motion of the slab is stopped. For weak slabs that lie upon the lower mantle the horizontal motion is diminished, but not entirely stopped, due to traction at the upper/lower

mantle boundary. In models with no lower mantle, the cessation of horizontal motion is more complete and immediate owing to the no slip boundary condition.

## 4.5   Steady-state Subduction

The concept of steady-state subduction as invoked in scientific literature is variously defined depending on the context of a particular paper. Articles that examine earthquake cycles consider the long term motion of the plates to be steady-state as

compared to the stick and slip of earthquakes that take place on time scales of decades or centuries (Savage, 1983; Fukahata et al., 1996; Wang et al., 2012). Studies that examine the state of subduction zones in terms of heat Molnar and England (1995); Currie (2004), deformation (Sato and Matsu'ura, 1988; Fukahata et al., 1996; Wang et al., 2012) or asthenosphere flow (Funiciello et al., 2006) exclude the deeper mantle and interaction of the subducted slab and the lower mantle. Further instantaneous single time-step models are necessarily steady-state. Several studies, including this one, suggest that subduction

on geologic timescales is not steady-state. Models that assimilate realistic plate history including changing plate velocities and plate boundaries generate slab morphologies that are comparable to tomographic observations (Liu and Stegman, 2011). The analogue models of Guillaume et al. (2009) result in systems with dynamic trench migration and "never [reach] any steady-state regime" with dip angle oscillating between steepening and flattening at the surface as the slab buckles and folds at the bottom of the upper mantle. Zhong and Gurnis (1995) subduction models exhibit non-steady state changes to plate velocity

as subducted slabs entered the lower mantle. In work by Stegman et al. (2006) and Bellahsen (2005), steady state is referred to in the context of a "steady state phase." In this study we find that the state of subduction zone is in change throughout the span of the experiment. The primary metric of radius of curvature we note is firstly noisy throughout the model run. This is true for any method used to calculate the radius of curvature. For our preferred method, fitting a circle to 150 km, the change in radius of curvature exceeds the noise and captures salient behavior as the model evolves. Work by Capitanio et al. (2009)

finds that curvature is modified to minimize bending dissipation. In the models of this study a decrease in radius of curvature is associated with the relaxation from the initial condition and the descent of the slab through the upper mantle. An increase in the rate of radius reduction begins as the slab encounters the lower mantle and continues as the slab lies upon the upper/lower mantle boundary. The radius of curvature increases and decreases in a manner similar to Guillaume et al. (2009) during slab buckling.

## 5   Conclusions


Our evaluation of three methods for calculating radius of curvature showed that its time varying nature is related to the state of the subducted slab. The "spline" method is subject to variability in models due to its nature and selection of a smoothing



parameter. The "spline" method results in a smaller radius of curvature than either of the two "circle" methods. The circle methods are sensitive to changes of the morphology of the subducted slab, while the "spline" method is not. The method of fitting a circle to slab shape from the surface to a depth of 150 km is sensitive to the state of the subducted slab below 150 km, covers a wide range of radii of curvature, and is less noisy than other methods.

The strength of plates and slabs independently control the shape of the shallow slab and subducted slab. The richness of slab morphology as seen in tomographic inversions suggests that the mechanical strength of tectonic plates is modified by some mechanism. As in Petersen et al. (2015), we find that too weak surface plates transition from single-sided occurs early in the model evolution. That study also found that plates that are too strong can't be bent and subducted, resulting in the system evolving into the stagnant-lid regime.

Strong plates resist bending and preserve larger radii of curvature. Smaller radii of curvature a result of vertical descending or steeply dipping slabs, promote 2-sided convection. Larger radii of curvature preserve single-sided subduction. At depth strong slabs embed in the lower mantle, transmit bending stresses to the surface and decrease radii of curvature. Weak slabs at depth are less coupled to the lower mantle, and inhibit the transmission of stress to the surface.

"Earth like" subduction is single-sided, supports wide range of radii of curvature and slab morphologies. In this study the
richness of such observations is best reproduced by a combination of strong surface plates and weakened slabs. The strong plates promote single-sided subduction while the weakened slabs allow slabs to bend, pile or lie flat, as is seen in tomographic inversions.

*Acknowledgements.* This material is based upon work supported by the National Science Foundation under Grant No. 1255040. This work used the Extreme Science and Engineering Discovery Environment (XSEDE), which is supported by National Science Foundation grant
number ACI-1053575. The authors acknowledge the Texas Advanced Computing Center (TACC) at The University of Texas at Austin for providing HPC resources that have contributed to the research results reported within this paper.



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





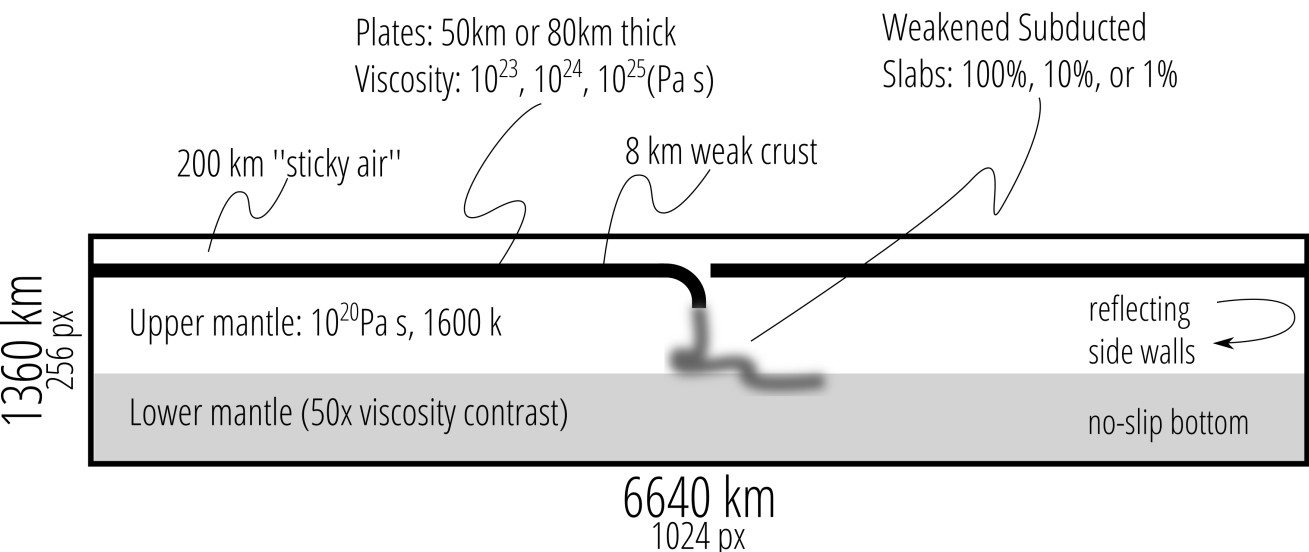

**Figure 1.** Diagram of the common parameter space, initial, and boundary conditions for models.

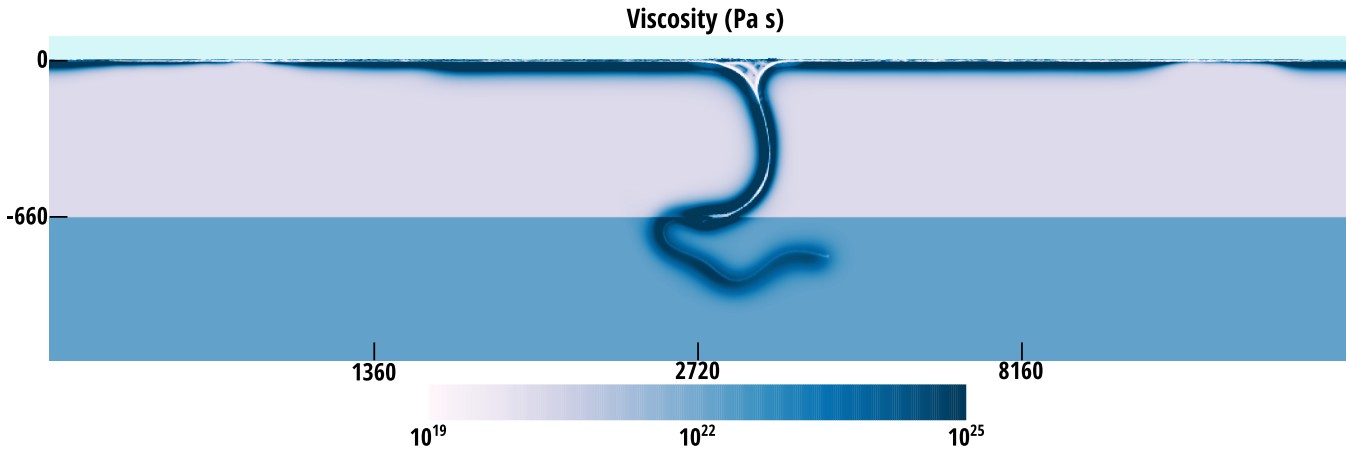

**Figure 2.** Piled slab descending into the lower mantle.





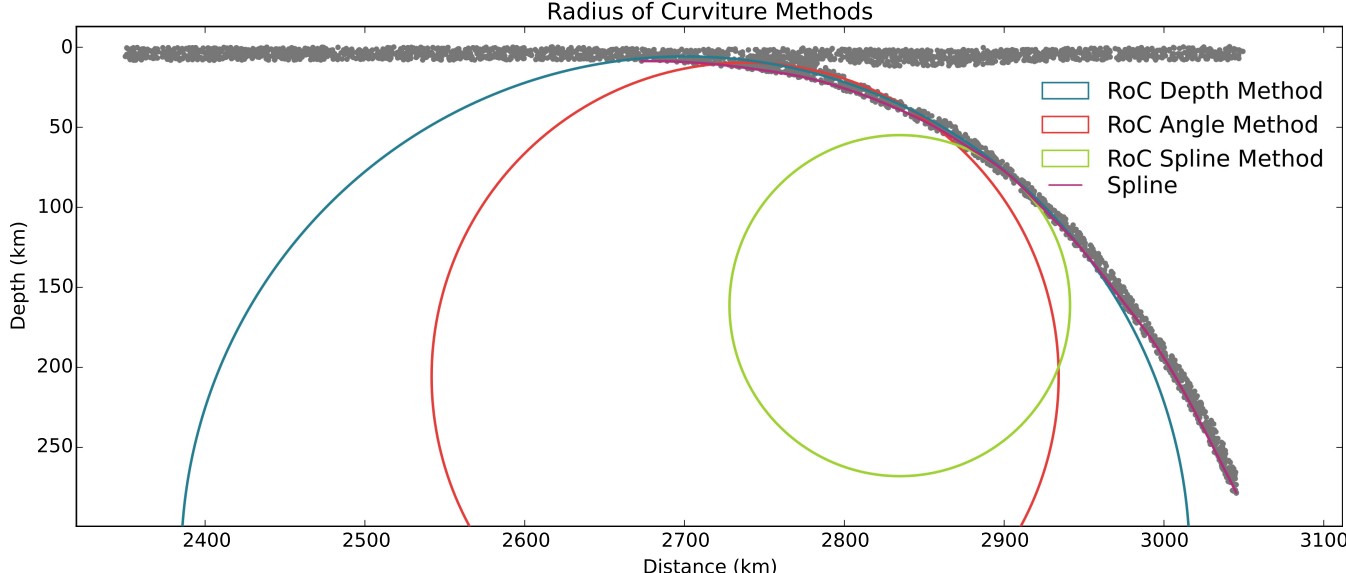

**Figure 3.** A comparison of the radius of curvature calculated for each of the methods tested for model 14.



**Figure 4.** Radius of curvature over time for each method tested for model 14.





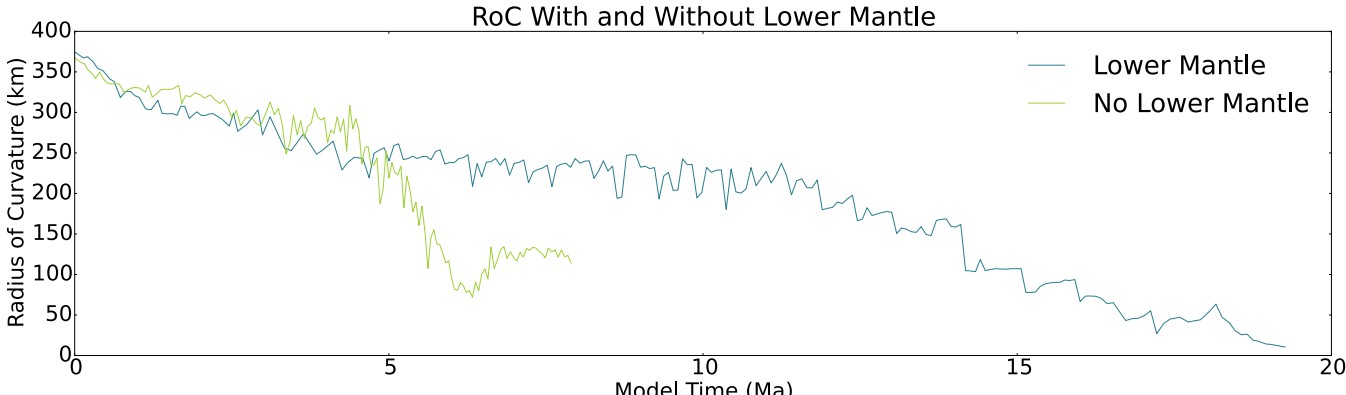

**Figure 5.** The radius of curvature over time for models 18 and 36, each with the same parameters except for the presence of a lower mantle.

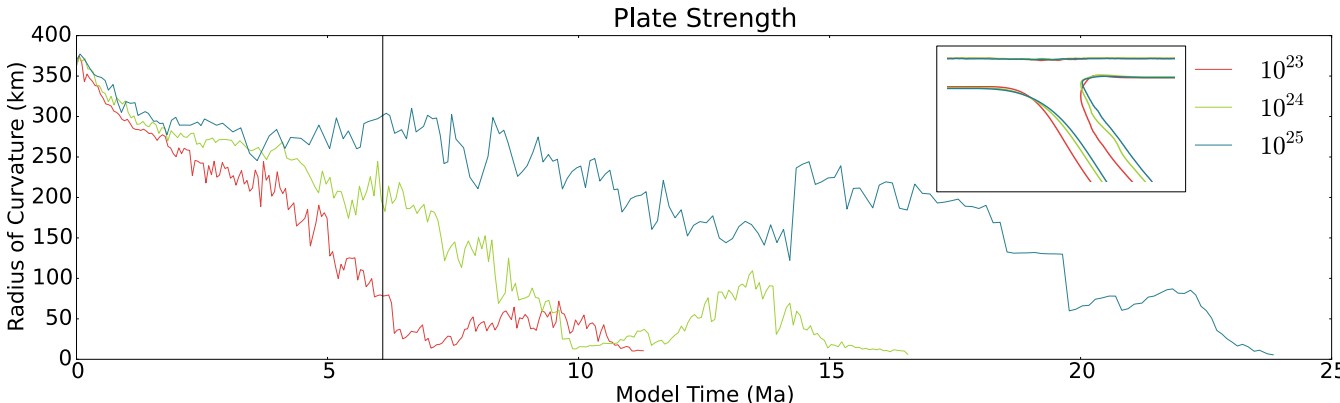

**Figure 6.** The change in radius of curvature over time for three models of different plate strength, models 11, 14, and 17. The inset shows the 1550K isotherm at the plate interface.



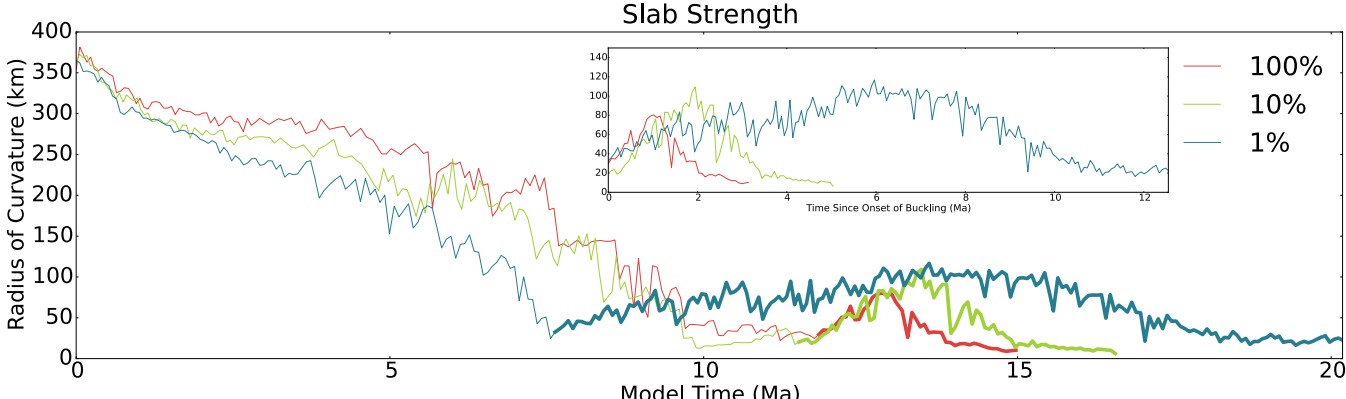

**Figure 7.** The change in radius of curvature for three models with the same plate strength and three different weakening parameters, models 15, 14, and 13. Bold lines show the period of slab buckling. The inset shows the radius of curvature with the plot shifted to the onset of buckling.

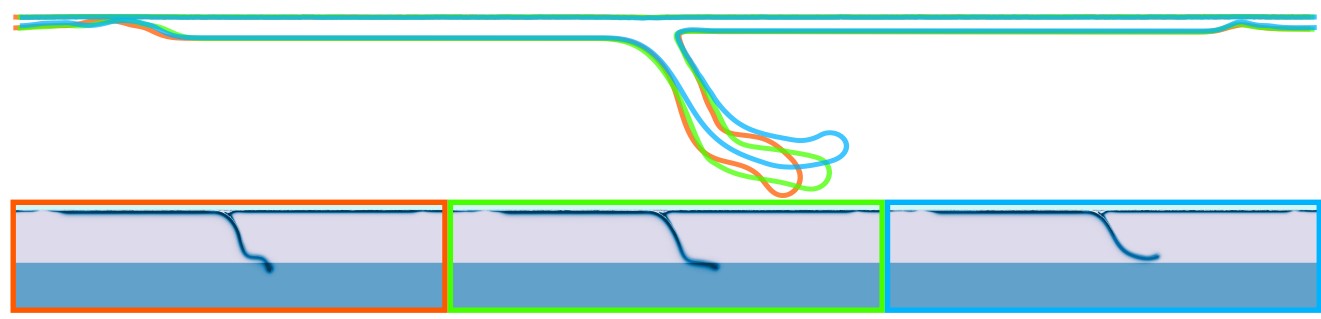

**Figure 8.** 1550 K isotherm for models 18 (left/orange), 17 (center/green), and 16 (right/blue). Surface plates are 80 km thick and have a maximum viscosity $10^{25}$ Pa s, the slab strength for models 18, 17, and 16 is $10^{25}$ Pa s, $10^{24}$ Pa s, and $10^{23}$ Pa s respectively. Subfigures show viscosity for each model.




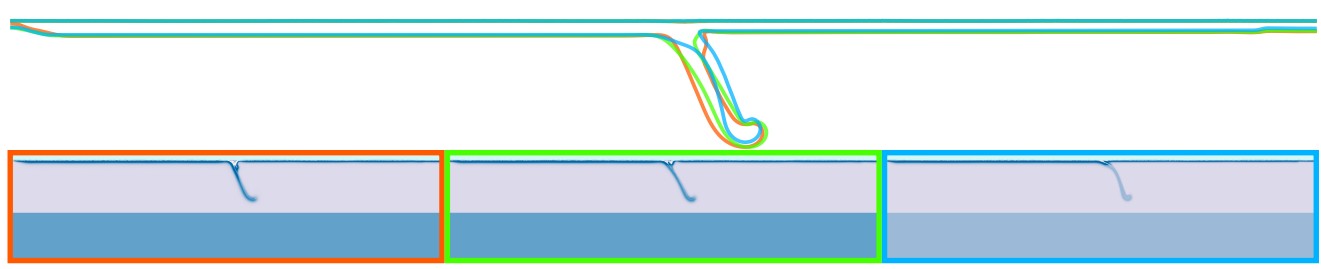

**Figure 9.** 1550 K isotherm for models 3 (left/orange), 2 (center/green), and 1 (right/blue). Surface plates are 50 km thick and have a maximum viscosity $10^{23}$ Pa s, the slab strength for models 3, 2, and 1 is $10^{23}$ Pa s, $10^{22}$ Pa s, and $10^{21}$ Pa s respectively. Subfigures show viscosity for each model.

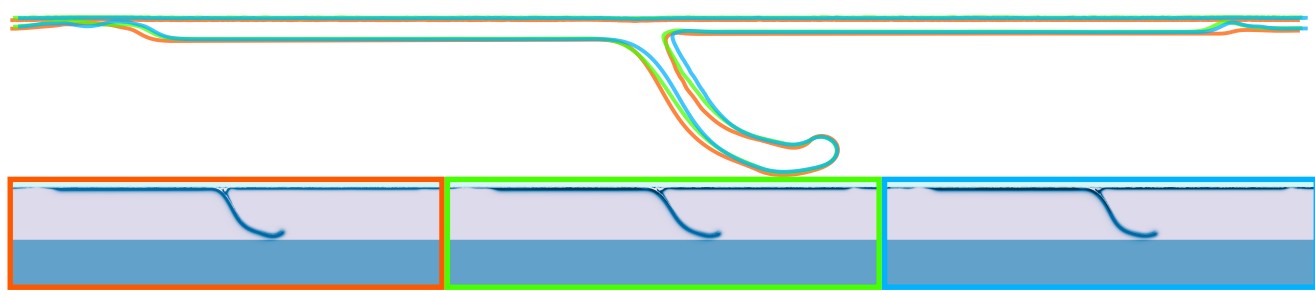

**Figure 10.** 1550 K isotherm for models 12 (left/orange), 14 (center/green), and 16 (right/blue). Surface are plates are 80 km think and have a maximum viscosity of $10^{23}$ Pa s, $10^{24}$ Pa s, and $10^{25}$ Pa s for models 12, 14, and 16 respectively. Slabs have a maximum viscosity of $10^{23}$ Pa s. Subfigures show viscosity for each model.

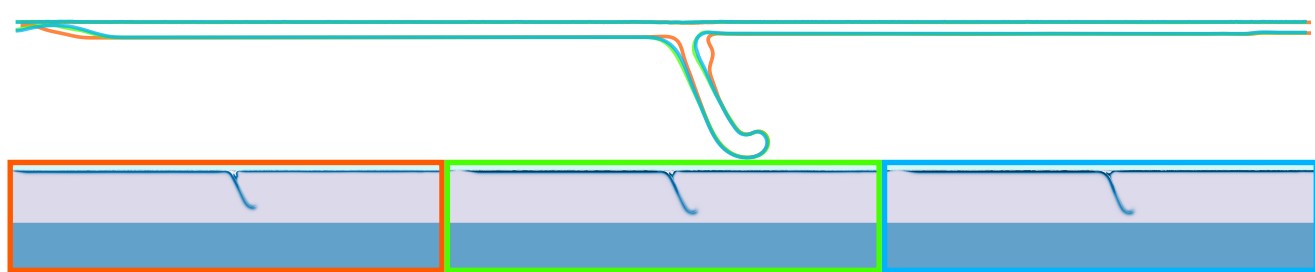

**Figure 11.** 1550 K isotherm for models 3 (left/orange), 5 (center/green), and 7 (right/blue). Surface plates are 50 km think and have a maximum viscosity of $10^{23}$ Pa s, $10^{24}$ Pa s, and $10^{25}$ Pa s for models 3, 5, and 7 respectively. Slabs have a maximum viscosity of $10^{23}$ Pa s. Subfigures show viscosity for each model.







| Parameter | Description | Value |
|---|---|---|
| $x$ | Domain width | 5440 km |
| $z$ | Domain depth | 1360 km |
| $nx$ | Horizontal cells | 1024 |
| $nz$ | Vertical cells | 256 |
| $g$ | Gravitational acceleration | $9.81\ \mathrm{ms}^{-2}$ |
| $\rho_0$ | Reference density | $3300\ \mathrm{kgm}^{-3}$ |
| $Cp$ | Heat capacity at constant pressure | $1200.0\ \mathrm{J\ K}^{-1}$ |
| $k$ | Thermal conductivity | $3\ \mathrm{Wm}^{-1}\mathrm{K}^{-1}$ |
| $\eta_{ref}$ | Reference viscosity at $T = 1600\ K$ | $1 \times 10^{20}\,\mathrm{Pas}$ |
| $\eta_{air}$ | Viscosity of "sticky air" | $1 \times 10^{18}\,\mathrm{Pas}$ |
| $h_{air}$ | Air layer thickness | 100 km |
| $h_{weak\ crust}$ | Weak crustal layer thickness | 8 km |
| $C_{mantle}$ | Mantle cohesion | 150.0 MPa |
| $\mu_{mantle}$ | Mantle coefficient of friction | 0.1 |
| $C_{weak\ crust}$ | Weak crust cohesion | 1.0 MPa |
| $\mu_{weak\ crust}$ | Weak crust coefficient of friction | 0.0 |

**Table 1.** Parameters common to all models



| Model # | Max. Viscosity (Pa s) | Lithosphere Thickness (km) | Slab Strength (% of Max. Viscosity) | Lower/Upper Mantle Contrast |
|---|---|---|---|---|
| 1 | $10^{23}$ | 50 | 1 | 50 |
| 2 | $10^{23}$ | 50 | 10 | 50 |
| 3 | $10^{23}$ | 50 | 100 | 50 |
| 4 | $10^{24}$ | 50 | 1 | 50 |
| 5 | $10^{24}$ | 50 | 10 | 50 |
| 6 | $10^{24}$ | 50 | 100 | 50 |
| 7 | $10^{25}$ | 50 | 1 | 50 |
| 8 | $10^{25}$ | 50 | 10 | 50 |
| 9 | $10^{25}$ | 50 | 100 | 50 |
| 10 | $10^{23}$ | 80 | 1 | 50 |
| 11 | $10^{23}$ | 80 | 10 | 50 |
| 12 | $10^{23}$ | 80 | 100 | 50 |
| 13 | $10^{24}$ | 80 | 1 | 50 |
| 14 | $10^{24}$ | 80 | 10 | 50 |
| 15 | $10^{24}$ | 80 | 100 | 50 |
| 16 | $10^{25}$ | 80 | 1 | 50 |
| 17 | $10^{25}$ | 80 | 10 | 50 |
| 18 | $10^{25}$ | 80 | 100 | 50 |
| 19 | $10^{23}$ | 50 | 1 | 1 |
| 20 | $10^{23}$ | 50 | 10 | 1 |
| 21 | $10^{23}$ | 50 | 100 | 1 |
| 22 | $10^{24}$ | 50 | 1 | 1 |
| 23 | $10^{24}$ | 50 | 10 | 1 |
| 24 | $10^{24}$ | 50 | 100 | 1 |
| 25 | $10^{25}$ | 50 | 1 | 1 |
| 26 | $10^{25}$ | 50 | 10 | 1 |
| 27 | $10^{25}$ | 50 | 100 | 1 |
| 28 | $10^{23}$ | 80 | 1 | 1 |
| 29 | $10^{23}$ | 80 | 10 | 1 |
| 30 | $10^{23}$ | 80 | 100 | 1 |
| 31 | $10^{24}$ | 80 | 1 | 1 |
| 32 | $10^{24}$ | 80 | 10 | 1 |
| 33 | $10^{24}$ | 80 | 100 | 1 |

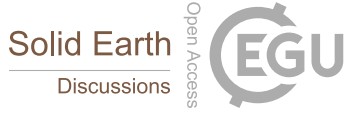

| Model # | Max. Viscosity (Pa s) | Lithosphere Thickness (km) | Slab Strength (% of Max. Viscosity) | Lower/Upper Mantle Contrast |
|---------|------------|------------|------------|------------|
| 34 | $10^{25}$ | 80 | 1 | 1 |
| 35 | $10^{25}$ | 80 | 10 | 1 |
| 36 | $10^{25}$ | 80 | 100 | 1 |

**Table 2.** Model Parameters