# Peer review of "The Subduction Dichotomy of Strong Plates and Weak Slabs"

_Solid Earth, 2016_

## Referee Comment (RC1) · Anonymous Referee #1 · 26 Apr 2016

In this article the authors have modelled 2D subduction by varying the strength of both the plate and the slab segments. The major finding is that, according to the authors, most observations at subduction zones are reproduced for a combination of strong plates and weak slabs.

However, I find that, as it is, the article presents some physical assumptions which are difficult to reconcile with the nature: 1) first of all, slab weakening is imposed 10 km below the base of the horizontal plates. As far as I know, there is no diffuse weakening mechanisms occuring at these depths, with the exception of slab dehydration that on the one side can induce dehydration embrittlements, but on the other side dries up the slab and make it stronger after the fluids have percolated away from it. Another weakening mechanism often invoked is grain-size reduction due to phase transformation, but this occur around the transition zone, that is much deeper than 60 or 90 km used

in the calculations. 2) could the authors explain why a mantle cohesion = 150 MPa is needed in the Byerlee's law?

The authors state that a weak slab is needed to reconcile models with observations. In my opinion this is not entirely true, as tomographic image from Tonga, for example, show that what appear the tip of the slab is dipping into the lower mantle (similar to their model 18). And there we know that the slab must be very old and stiff because of the very fast subduction rate.

In general, I feel that the conclusions relative to the role of the rheology of subducting plates are too superficial and not strongly supported by observations.

However, I have really appreciated the part of the article where different fitting methods used to measure the plate curvature were investigated, and which among them gives the better estimates and the sense of the non-steady state subduction. Hence, in conclusion, I would suggest to reshape the article toward this part of the results, emphasizing also how other studies using the spline method may have wrongly estimated the plates' radius of curvature.

---

## Referee Comment (RC2) · B.J.P. Kaus (Referee) · 26 May 2016

The manuscript discusses the effect of having slabs that are weak at depth but strong close to the surface on the dynamics of slab bending and subduction. Even though there are a few interesting aspects to the work (such as comparing various methods to compute the radius of curvature, and the temporal evolution of these parameters), I have a number of more serious concerns that should be addressed:

1) One of the main points of the MS seems to be to argue that slabs are strong at the surface and become very weak at depth. This depth-weakening is achieved by lowering the maximum cutoff viscosity at depth. The authors vaguely argue that this:

"is a computational efficient way of parameterizing the effective viscosity that may result due to any number of mechanisms such as Peierls creep, damage rheology or other

Non-Newtonian rheologies" [p. 2]

I disagree; what you are doing is rather arbitrary and has little to no physical justification. If you lower the maximum cutoff viscosity, you reduce the plate strength everywhere, and importantly the viscosity contrast between the slab & surrounding mantle (which causes the buckling effects you are after). This is in my opinion very different than the potential mechanisms you put forward:

a) Peierls plasticity has a slight depth dependence which lowers the plate strength only in locations where the stresses are reaching the yield stress (which is on the order of several 100 MPa for most Peierls plasticity models), In my experience this is something that occurs throughout the bending zone, but certainly not at much larger depths such as in the mid upper mantle or close to the 660 (even though exceptions can occur in the bending zone around the 660).

b) Non-Newtonian viscosities depend on the local strainrate, and this strainrate is quite similar outside and inside the slab (at least in most models I performed sofar). The result of that is that you weaken the asthenosphere in pretty much the same manner as the slab itself and therefore there is very little difference in the effective viscosity contrast between the slab and the surrounding mantle (and the slab is thus not weakened in the manner that you describe).

c) There might ofcourse be a damage rheology that does exactly what you simulate, but I am not familiar with what this mechanism should be (it would be good to have experimental evidence for it as well).

Given how crucial this assumption is for your manuscript, this should be thoroughly and convincingly discussed (and requires certainly more text). Alternatively, you instead focus on the effects of slab curvature and it's temporal evolution, and change the title and text accordingly (which seems to be the more natural thing to do).

2) Overall, the MS seems to be more on how slab strength controls the bending radius

and how this changes with time, rather than on the dichotomy itself or on why this is the preferred model for Earth. As far as I understand, whether a slab really buckles at the 660 or not is poorly constrained from tomographic models, which can at max. see a pile of slabs but not how those are internally deformed. Earlier work (e.g. Ribe et al. 2007 EPSL) has shown similar folding (among many other papers) and discussed that the thicker high velocity zones above the 660 might be explained by such a folding mechanism. Yet, the emphasis is on "might", and it is far from clear whether this is indeed the case. Using it as a fact, as you do in the conclusions, and taking this as proof that slabs must therefore be weakened at depth by "some mechanism [conclusions]" is not a very strong argument in my opinion.

More minor issues:

- figure 1: you write that the sticky air layer is 200 km thick, but in the text & table it is 100 km. - page 4, line 13/14: maybe rephrase to ".. lower mantle boundary, after which it buckles and piles" - page 4, line 20: You prescribe a slab with a radius of curvature of 400 km, and none of your methods is capable of recovering this radius of curvature? To me it seems that there is something iffy here. Have you tested that your methods are implemented correctly? It certainly deserves more discussion. - page 6, line 30: "have the advantage that the" - section 4.4: I find it encouraging that there is actually very little difference between models with a no slip lower BC and a highly viscous lower mantle at least during the initial model stages. As the viscosity jump effectively acts as a no-slip or a difficult-to-slip boundary this is not entirely surprisingly. The speedup of the slab that you discuss in the text is not apparent from figure 5, so maybe you can add insets of how the slab morphology looks like after a predefined time? - figure 3: Radius of curvAture

Boris Kaus, Mainz

---

## Author Comment (AC1) · 21 Jun 2016

Referee #1 enumerates 2 points to address.

The first that slab weakening is imposed at the, too shallow, depth of 10km below the overriding plate. Our paper presents two figures which compare two sets of models that address this concern. Figure 8 show three models with equal strength 80km thick plates and maximum slab viscosity of 1e25, 1e24, 1e24 Pa s. The figure shows that from the surface to a depth to several hundred kilometers the slabs have a substantially similar morphology. Contrastingly figure 9 shows three models in which the strength of the 80km thick plates is varied while the slab strength at depth is equal across models. The maximum viscosity of the surface plates in 1e23, 1e24,and 1e25 and the subducted slab strength is 1e23. In this figure the the morphology of the subducted

slabs differ in the shallow part of the mantle (to several 100km) while the portion of the slab lying above the lower mantle across all models is substantially similar. We would assert that the shape of subducted slabs in the shallow upper mantle is controlled by plate strength and the shape of the subducted slab deep into the upper mantle is controlled by the subducted slab strength if not fully independently then at least to a very high degree. While we believe that it would be an interesting exercises to systematically modify the depth of weakening to discover the degree to which these mechanisms are, in fact, independent the results of this study do not suffer due to the chosen depth of weakening.

The second point raised by the referee questions the cohesion parameter of 150 MPa. The use of 150MPa limits yielding below the crustal layer increasing plate strength. As our models do not use an imposed weak zone, nor do they limit the weak crust to the subducting plate, as is found in other studies, the high cohesion serves to strengthen the overriding plate. Yielding is focused in the hinge of the subducting plate. This is can be seen in the figures that plot viscosity, e.g. figure 8, figure 10 models 14 and 16, and figure 11. The value that we use for this parameter, 150 MPa, is the same as that used by other studies that generate a distribution and fragmentation of tectonic plates similar to Earth (Mallard et al., Nature 2016).

The referee compares our model 18, a model with no weakening, to tomographic observations below Tonga, suggesting that no weakening in required to support observations. Pysklywec et al, EPSL 2003 (doi:10.1016/S0012-821X(03)00073-6), using the tomography model of van der Hilst et al., Nature 1997 (doi:10.1038/386578a0), model an avalanche scenario for the Tonga slab using an initial condition that is similar to the evolved state of model 18 as shown in figure 8. In other regions tomographic interpretations suggest piling or buckling or slabs. Under the Izu-Bonin and Mariana convergent zones van der Hilst and Seno, EPSL 1993 (doi:10.1016/0012-821X(93)90253-6), have suggested that the subducted plate has piled. Ribe et al., EPSL 2007 (doi:10.1016/j.epsl.2006.11.028), draw comparisons between analytic and

numeric models of viscous sheet buckling instabilities and tomographic models of this Cocos plate beneath Central America, and of the Australian plate beneath Java. They identify large wedge-like structures which they interpret as "'piles' of folds generated by buckling instabilities in the transition zone". Therefore, the same strength of slab that can match observations of Tonga cannot match observations of Marianas, and this is why we model systems with a weakening mechanism with a range of various magnitudes. It is this range that gives rise to models that behave as both "strong" slab systems comparable to the Tonga trench and to "weak" slabs systems that buckle and pile comparable to the Cocos, Australian, Izu-Bonin and Mariana subduction systems.

---

## Author Comment (AC2) · 21 Jun 2016

Referee #2 enumerates 2 major points to address.

The first point is regarding the method used to weaken slabs at depth. As described in the text we weaken slabs using a parameterization that limits the maximum viscosity at depth. The rheological laws, and parameters of the same, have been determined empirically from laboratory experiments on Earth materials using sample sizes millimeters or centimeters in size and at strain rates several order of magnitude larger than deformation in tectonic settings. A common practice in computational modeling is to adopt specific values for parameters from these laboratory-derived empirical relationships and employ them in models of regional or global extent, but formally applied at length-scales and timescales determined by the resolution of the computational mesh in the

numerical model. This process requires some level of arbitrary decision making, and is reinforced by the modeler's experience and understanding of how these parameters change the overall behaviour of the system. In much the same tradition the weakening mechanism employed in our models achieves the overall effect of diminishing the mesoscale strength of the plate, on the order of tens of kilometers and on timescales of thousands of years and larger, and this scale is not represented by the empirical relationships derived from laboratory experiments. The deformation at the mesoscale includes subgrid physics and phenomena including an undetermined number of unresolved faults that allow for deformation and fluid transport, both of which will enhance the weakening present in typical flow laws developed from cm size samples.

Furthermore, in models that use a yield stress for modeling pseudoplasticity, it is common practice to assume that the material strength is instantaneously renewed back to its full strength, as determined by the empirically-derived flow laws, as soon as it is no longer subjected to stresses that exceed the yield stress. Although this is typically done, this represents an implicit assumption being made in such models. However, materials that have undergone significant yielding have likely developed other weaknesses (e.g. damage) at both the grain scale as well as the mesoscale including tectonic fabrics and generation of subgrid faults. Without labeling the decisions as whether to include, or not include, a post-yielded reduction in material strength, as arbitrary, our models represent an alternative to the defacto assumptions that are implicitly made. The interpretation of rheological relationships in numerical modeling outside of the experimental conditions that they are based upon is an important scientific discussion that should be held in the community, and at this point there are several alternatives that justify further exploration.

The referee has also raised the point that the mechanisms suggested in the manuscript, Peierls creep and Non-Newtonian creep, may not be present where we apply weakening. In a study by Garel et al, 2014, the authors developed models which used a composite viscosity calculated as the harmonic mean of several rheological

laws, Peierls and Non-Newtonian flow among them. In that study the authors find the primary deformation mechanism of Peierls creep to be widely active inside the slab to depths of 660. Further, they show that dislocation creep to be the primary deformation mechanism within the slab at the edges while there continues to be an overall viscosity contrast between slab and background mantle. We have updated the Methods section to explicitly call out these results.

The second major point questions the interpretation of slab piles. The referee suggests that our interpretation of slab piles at the upper mantle/lower mantle boundary is stronger than the literature would allow. The referee cites Ribe 2007 stating that in that manuscript the interpretation is that the piles "might" be folded slabs. That manuscript addresses the "principal alternative" to folded slabs of shear thickening and citing Gurnis and Hager 1986 and Gaherty and Hager 1994, and concludes that the alternative mechanism is insufficient to reproduce what is seen in tomography: "The principal alternative, sometimes called 'advective' or 'pure shear' thickening, is uniform widening of the slab in response to the downdip compressional stress due to an increasing viscous resistance with depth. However, numerical models [24] and [9] show that this mechanism can thicken the slab by at most a factor of two, which is not sufficient to account for the tomographic observations." The reviewer states that, in our conclusions, we are taking the interpretation of piles as folded slabs as "a fact." Our conclusion is that in these models a weakening mechanism allows for folded slabs and, as compared to models without weakening, best reproduces those types images seen in tomography. Our discussion section has been updated to acknowledge that the interior structure of the observed piles is unknown and to cite Ribe's discussion of the likelihood of alternatives.

Referee 2 minor issues:

"figure 1: you write that the sticky air layer is 200 km thick, but in the text & table it is 100 km." The figure has been updated to reflect a 100km thick layer of sticky air

"page 4, line 13/14: maybe rephrase to '.. lower mantle boundary, after which it buckles and piles'" Updated text accordingly.

"page 4, line 20: You prescribe a slab with a radius of curvature of 400 km, and none of your methods is capable of recovering this radius of curvature? To me it seems that there is something iffy here. Have you tested that your methods are implemented correctly? It certainly deserves more discussion."

The fact that no method returns exactly the prescribed radius of curvature is a result of three factors: first, the 400km radius of curvature used in the initial condition is at the surface of the plate while the fit circles are mid-crust, second, the least squares fit to a cloud of randomly placed points is a statistical solution subject to noise, third, and most significant, no method selects all of the points from the trench to the tip of the slab. This "angle" method, in the case of a perfect circle will select points down to a depth of 117 km. The "depth" method is prescribed to select points no deeper than 150km. The "spline" method is also a statistical method, subject to an arbitrary smoothing parameter and the radius of curvature at any given point is only sensitive to nearby particles. Both of the circle methods only use an arclength that is a fraction of the total circle, which leads to the underestimation of the radius of curvature, and using a longer arclength recovers the correct radius of curvature. However, for this study we use an arclength that is approximately equal to length from the trench to the depth of the seismogenic zone.

The above explanation has been added to the text in the Radius of Curvature subsection of the Results Section. We have also added text in the initial condition section that makes it clear the radius of curvature is from the surface of the slab. Given a set of points that lie exactly on a circle each of the circle methods does return precisely the correct answer. The spline method is nonetheless subject to the smoothing parameter.

- page 6, line 30: "have the advantage that the" Updated text accordingly.

- section 4.4: I find it encouraging that there is actually very little difference between

models with a no slip lower BC and a highly viscous lower mantle at least during the initial model stages. As the viscosity jump effectively acts as a no-slip or a difficult-to-slip boundary this is not entirely surprisingly. The speedup of the slab that you discuss in the text is not apparent from figure 5, so maybe you can add insets of how the slab morphology looks like after a predefined time? An inset showing both models at 4.4Ma is included in that figure and the caption has been updated to reflect this.

- figure 3: Radius of curvAture Figure title fixed accordingly.